# Doped N/Ag Carbon Dot Catalytic Amplification SERS Strategy for Acetamiprid Coupled Aptamer with 3,3′-Dimethylbiphenyl-4,4′-diamine Oxidizing Reaction

**DOI:** 10.3390/nano9030480

**Published:** 2019-03-25

**Authors:** Xiaozhen Feng, Chongning Li, Aihui Liang, Yanghe Luo, Zhiliang Jiang

**Affiliations:** 1Key Laboratory of Ecology of Rare and Endangered Species and Environmental Protection (Guangxi Normal University), Ministry of Education, Guilin 541004, China; fxz97118@guet.edu.cn (X.F.); lcn7882342@163.com (C.L.); ahliang2008@163.com (A.L.); 2School of Life and Environmental Sciences, Guilin University of Electronic Technology, Guilin 541004, China; 3School of Food and Bioengineering, Hezhou University, Hezhou 542899, China

**Keywords:** acetamiprid, aptamer, carbon dot catalytic, DBD, SERS

## Abstract

The as-prepared co-doped N/Ag carbon dot (CD_NAg_) has strong catalysis of H_2_O_2_ oxidation of 3,3′-dimethylbiphenyl-4,4′-diamine (DBD). It forms an oxidation product (DBD_ox_) with surface-enhanced Raman scattering (SERS) activity at 1605 cm^−1^ in the silver nanosol substrate, and a CD_NAg_ catalytic amplification with SERS analytical platform can be structured based on aptamer (Apt) with the DBD oxidizing reaction. For example, the aptamer (Apt) of acetamiprid (ACT) can be adsorbed on the surface of CD_NAg_, resulting in inhibited catalytic activity, the reduced generation of DBD_ox_, and a weakened SERS intensity. When the target molecule ACT was added, it formed a stable Apt-ACT complex and free CD_NAg_ that restored catalytic activity and linearly enhanced the SERS signal. Based on this, we proposed a new quantitative SERS analysis method for the determination of 0.01–1.5 μg ACT with a detection limit of 0.006 μg/L.

## 1. Introduction

Exploring highly selective, sensitive, simple, and rapid contaminant analysis methods is one of the vital development directions of environmental analytical chemistry. Highly selective and sensitive assays rely on highly selective analytical reactions and highly sensitive detection techniques. Nucleic acid aptamers (Apt) are oligonucleotide fragments that have specific recognition of and high affinity with the target. It has applications in genomics, food safety, medical diagnosis, and biomedicine [1,2,3,4,5,6,7,8,9,10,11,12,13]. Surface-enhanced Raman scattering (SERS) is a sensitive molecular spectral technique with real-time in-situ detection; less sample is needed, there is little damage to the sample, and it is simple and fast [5,6,7,8,9,10]. Although many SERS studies have been reported, most of them were qualitative due to poor reproducibility. Reproducibility not only affects qualitative analysis but is a principle of quantitative analysis, which limits its development. To overcome the shortage, some reproducible preparations of clean and highly Ag and Au SERS active substrates have been reported in Reference [14]. Bassi et al. [15] prepared recyclable SERS active glass chips. The gold nanostars were grafted on functionalized glasses by means of electrostatic interactions and then they were coated with a silica layer of controllable thickness. The SERS activity was examined in terms of reproducibility using rhodamine 6G molecular probes. As an analytical technique, the highly sensitive and selective SERS quantitative analytical method is important. It depends on reproducible SERS substrates and highly selective biochemical reactions such as Apt reactions. Combining the high selectivity of the Apt reaction with the high-sensitivity of SERS detection to develop a SERS quantitative analytical method has achieved favorable results. Xu et al. [3] established a simple, selective and sensitive SERS aptasensor to detect 10^2^ to 10^7^ cfu/mL Salmonella typhimurium in milk samples, based on gold nanodimers assembled with a Raman signal and capture probe via hybridization of the complementary ssDNAs. Song et al. [4] proposed an approach based on SERS and As(III)-Apt for the detection of 0.5 to 10 ppb As(III) with a detection limit of 0.1 ppb. Wen et al. [11] described Apt modified nanosilver to obtain a nanosilver-Apt SERS probe for the determination of 6.3–403.6 μg/L melamine. Ye et al. [12] introduced nanogold modified by Apt to be used in the preparation of nanogold-Apt probes for 0.29–23.0 ng/mL As^3+^, with a detection limit of 0.1 ng/mL. Dong et al. [13] developed a nanosenor to detect 1.0 pg/mL–10 ng/mL melamine in milk using an aptamer-modified SERS nanosensor and oligonucleotide chip. However, there are few reports on the Apt-mediated CD_NAg_ nanocatalytic amplification of SERS probes for acetamiprid (ACT).

As a new type of fluorescent nanomaterial, carbon dots (CDs) have attracted great attention, due to their excellent optical properties, satisfactory chemical stability, low toxicity, exceptional biocompatibility and surface functional adjustability. They have become the most popular carbon nanomaterials after fullerenes, carbon nanotubes and graphene, and have been studied and applied in bio-imaging and fluorescence sensing [16]. The preparation of CDs has been one of the hotspots in nanomaterials and analytical chemistry. At present, some synthesis methods for CDs have been established, such as arc discharge, laser etching, chemical oxidation, template methods, hydrothermal synthesis and microwave methods [17]. The microwave method has attracted much attention because of its fast preparation speed and harmless water as a solvent. Xiao et al. [18] synthesized nitrogen-doped CDs with strong photoluminescence by a one-step microwave irradiation of citric acid and ethylenediamine for 15 min, used for cell imaging. Water-soluble phosphorus-containing CDs with strong green fluorescence can be synthesized by one-step microwave irradiation using ethylenediamine as the nitrogen-doping agent and phosphorus-rich phytic acid as the carbon source [19]. The main analytical application of CDs is fluorescence analysis. Most of the methods are based on the redox reaction between the carbon point fluorescence probe and the analyte, a coordination reaction leading to fluorescence quenching or fluorescence enhancement [20]. Ahmed et al. [21] used ethylene glycol bis-(2-aminoethyl ether)-N,N′,N′,N′-tetraacetic acid and tri (hydroxymethyl) ethylenediamine as precursors to prepare CDs, by hydrothermal synthesis, for a long time. Based on the fluorescence quenched by 4-nitrophenol, a 28 nM 4-nitrophenol can be detected. The N-doped carbon dots were prepared rapidly by microwave, and used as the nanoprobe and nanocatalyst for the fluorescence determination of ultratrace isocarbophos with a label-free aptamer [22]. Luo et al. [23] used fluorescent CD-modified single-stranded nucleic acids and Apt-modified graphene oxide to undergo a chain substitution reaction, and 10–800 nM adenosine triphosphate were detected by fluorescence. Shi et al. [24] modified complementary nucleic acids with CDs as fluorescent labeling indicators. The Apts were fixed on the surface of Fe_3_O_4_ nanoparticles, and a fluorescence detection of 0.25–50 ng/mL beta-lactoglobulin was performed with a detection limit of 37 pg/mL.

ACT is a new high-efficiency broad-spectrum nicotine chloride insecticide. It can act on the synaptic nicotine acetylcholine receptors of the nervous system of homoptera, lepidoptera and coleoptera pests on vegetables, fruit trees, rice and tea. It interferes with the stimulation of and transmission in the nervous system of pests, blocks the nervous system pathways, and causes paralysis, thus achieving the insecticidal effect [25]. In natural environment, ACT is difficult to degrade under strong acidic and neutral conditions. It has been reported that low concentrations of ACT can impair long-term memory and sensitivity to stimulation in bees [26]. In addition, some studies have shown that ACT is also very harmful to the human body; it can induce DNA damage [27]. Therefore, it is necessary to strictly monitor the residues of ACT pesticides. At present, the main methods for detecting ACT are gas chromatography, high performance liquid chromatography, chromatography-mass spectrometry, enzyme-linked immunosorbent assay and electrochemical methods [28,29,30,31,32,33,34,35,36,37]. However, such methods are expensive, complex to operate or pretreat, have low sensitivity and are time-consuming. Therefore, it is necessary to develop a rapid, sensitive, and selective method for the determination of trace ACT pesticide residues. Jin et al. [33] reported a sensor for the detection of ACT in vegetables based on a photocatalytic degradation compound. In this paper, the Apt reaction [25] was coupled with the SERS catalytic reaction of 3,3′-dimethylbiphenyl-4,4′-diamine (DBD) oxidized product (DBD_ox_), and a novel, rapid and sensitive SERS method for the detection of ACT was established.

## 2. Experimental Section

### 2.1. Instruments

A model of DXR smart Raman spectrometer (Thermo, Waltham, MA, USA), with a laser wavelength of 633 nm, power of 3.5 mW, slit of 50 μm and acquisition time of 5 s, a model of a 3K-15 high-speed refrigerated centrifuge (Sigma Co., Darmstadt, Germany), a model of a 79-1 magnetic stirrer with heating (Zhongda Instrumental Plant, Nanjing, China), a model of an HH-S2 electric hot water bath (Earth Automation Instrument Plant, Jintan, China), a model of a WX-6000 microwave digestion instrument (Preekem Scientific Instruments Co., Ltd., Shanghai, China), a model of a two spectrum Fourier transform infrared spectrometer and its supporting tableting device (Bojin Elmer Co., Ltd., Shanghai, China), a model of an FD-1C-50 vacuum drying freezer (Hangzhou Jutong Electronics Co., Ltd., Hangzhou, China), and a model of a S-4800 field emission scanning electron microscope (Hitachi High-Technologies Corporation, Japan/Oxford company, Oxford, UK) were used.

### 2.2. Reagents

The aptamer sequence (Apt): 5′-CTGAC ACCAT ATTAT GAAGA -3′ (Shanghai Sangon Biotech Co., Ltd., Shanghai, China), acetamiprid (ACT) (98.5% purity, Shanghai Shenggong Biotechnology Co., Ltd.); imidacloprid (purity 98.5%, Shanghai Aladdin Biochemical Technology Co., Ltd., Shanghai, China); atrazine (purity 98%, Shanghai Aladdin Biochemical Technology Co., Ltd.); carbendazim (purity 99%, Shanghai Aladdin Biochemical Technology Co., Ltd.); and chlorpyrifos (purity 99%, Shanghai Aladdin Biochemical Technology Co., Ltd.) were used. A 10 mmol/L AgNO_3_ solution, 0.1 mol/L sodium citrate solution, 30% H_2_O_2_ solution, 0.1 mol/L NaBH_4_ solution, 0.2 mol/L pH 3.6 HAc-NaAc buffer solution, 1.0 mol/L HCl solution, and a 0.25 mol/L NaOH solution were prepared. A 30 mg/L ACT standard solution was prepared as follows: In a breaker, 7.5 mg ACT were dissolved in 20 mL anhydrous ethanol. It was transferred into a 250 mL volumetric flask and diluted to the mark with water. A 0.5 mmol/L DBD solution was prepared as follows: First, 11 mg DBD were dissolved and transferred to a 100 mL volumetric flask. This was diluted to the mark with the volume ratio of 6:4 ethanol:water. The nanosilver sol (AgNPs) was prepared as follows [38]: Water, 44 mL, was added to a conical flask. Then, 2 mL 10 mM AgNO_3_, 2.0 mL 100 mM trisodium citrate, 600 μL 30% H_2_O_2_, and 600 μL 0.1 M NaBH_4_ were added into the water successively. After the color turned blue, with rapidly stirring, the mixture was immediately shifted to the air light wave stove. After irradiation for 10 min at 250 °C, the color went from blue to orange-red. The mixture was cooled naturally to room temperature. It was then diluted to 50 mL. The concentration of the nanosilver sol was 0.4 mM Ag, calculated by AgNO_3_ concentration. All reagents were of analytical grade and the water was double distilled.

Preparation of CD_NAg_ was as follows: one gram of glucose, 0.8 g of urea and 0, 200, 500, and 700 μL 0.01 mol/L AgNO_3_ were sonicated in 30 mL water. Then, it was transferred to a high-pressure reaction vessel with a polytetrafluoroethylene substrate, sealed and irradiated in a microwave oven for 10 min with a power of 640 W. After the irradiation, it was cooled to room temperature to obtain a pale yellow solution, which was calculated by adding the total amount of carbon to obtain a solution of *C* = 13 mg/mL CD_N_, CD_NAg1_, CD_NAg2_ and CD_NAg3_, respectively.

### 2.3. Procedure

A suitable amount of 1.0 μg/L ACT standard solution, 25 μL of 1.55 μmol/L Apt, and 400 μL 1.7 mg/L CD, 60 μL 2 mmol/L H_2_O_2_, 75μL 0.5 mmol/L DBD, 100 μL 1 mmol/L pH 3.6 HAc-NaAc solution were added to a 5 mL test tube and then diluted to 1 mL with water and mixed well. The tube was placed in a bath at 50 °C for 20 min. The reaction was stopped by tap water cooling. A 400 μL 0.4 mmol/L AgNPs solution was added and diluted to 1.5 mL with water. The mixture was transferred to a quartz cell and its SERS spectra were recorded. The SERS peak intensity *I* at 1605 cm^−1^ was measured, the blank (*I*_0_) without ACT was recorded, and the Δ*I* = *I* − *I*_0_ was calculated.

## 3. Results and Discussions

### 3.1. Analytical Principle

In the pH 3.6 HAc-NaAc buffer solution, the CD had a strong catalytic effect on the reaction of H_2_O_2_-DBD to form DBD oxidation products (DBD_ox_). When the Apt was present, it adsorbed on the CD surface, resulting in a weakening of the catalytic action of CD. After the addition of the target molecule ACT, it specifically bound to the Apt and, with the CD released, the catalytic activity was restored. With the increase of ACT, there was more desorption of CD, a faster catalyzed H_2_O_2_-DBD reaction, and a greater the concentration of DBD_ox_ formed. After the addition of the AgNP nanosol substrate, the SERS signal enhanced linearly (Figure 1). Coupling this catalytic amplification reaction with the Apt reaction, a new SERS quantitative analytical method could be established for detecting ultratrace ACT.

### 3.2. SERS Spectra

In a pH 3.6 HAc-NaAc buffer solution, the H_2_O_2_-DBD system was difficult to react in the 60 °C water bath. In the presence of nano-catalysts such as CD_N_ and CD_NAg1–3_, the oxidation of DBD by H_2_O_2_ was catalyzed and the oxidation product of DBD_ox_ had the strongest SERS activity for the CD_NAg2_ catalytic system. The SERS substrates, such as AgNP, gold nanoparticles and CD_NAg_ sols, were examined. The AgNP was the most sensitive and was chosen for use. After the addition of AgNP sol as a SERS substrate, the CD_NAg2_ catalytic system exhibited a peak at 1189 cm^−1^, ascribing to the stretching vibration of C–N, at 1335 cm^−1^, ascribing to the bending vibration of C–H, and at 1402 cm^−1^, owing to the bending vibration of C=C. The strongest peak was at 1605 cm^−1^, ascribing to the tensile vibration of C=N and C=C–C=C (Figure 2a). When the Apt was added it was coated on the surface of the carbon dot, thereby inhibiting the catalytic activity of the CD, and the SERS signal attenuated (Figure 2b). With the addition of ACT, the combination of ACT and Apt specifically released CDs, which restored its catalytic activity. With the increase of the ACT concentration, the release of CDs increased. The SERS peak at 1605 cm^−1^ increased linearly, due to more DBD_ox_ products being generated (Figure 2c), and was chosen for use.

### 3.3. Nanocatalysis and Aptamer Inhibition

Under the selected conditions H_2_O_2_ and DBD systems were difficult to carry out. As the CD catalyst concentration increased, the catalytic ability enhanced, the DBD_ox_ increased, and the SERS signal increased linearly in the AgNP nanosol substrate. When Apt was added, the CDs were entrapped by the Apt, resulting in inhibition of the CDs catalysis, and the SERS peak intensity decreased linearly (Table 1). In this article, the slope of the linear relationship was used to measure the catalysis and inhibition with simplicity and rapidity. The CD_NAg2_ was strongest and the strongest inhibition was Apt-CD_NAg2_.

In the experiment, citric acid was used as the carbon source, urea was involved in the reaction, and a nitrogen source and AgNO_3_ as the Ag source were provided to produce a nitrogen/silver co-doped CD. The CD was composed of various molecules that contain amide chains and many N-Ag atoms on the surface. Due to the fact that N and Ag atoms are electron donating atoms, the CD with a proper amount of nitrogen and silver atoms containing a lone pair of electrons could enhance the π bond conjugation. The CD was a catalyst to speed up the redox electron-transfer of DBD and H_2_O_2_ to form DBD_ox_ with SERS activity. Nitrogen/silver-doped CDs showed high catalysis due to their greater number of surface electrons. This accelerated the redox electron transfer, which is a new type of CD catalyst with promising applications for the amplification of analytical signal.

### 3.4. Electron Microscopy (EM) and Infrared Spectra

According to the experimental method, 1.5 mL of the reaction solution were centrifuged in a 2 mL centrifuge tube for 10 min (70 × 100 rpm), the supernatant was discarded, and the volume was adjusted to 1.5 mL with water and sonicated for 15 min. We repeated the above centrifugation step twice and added 1.5 mL water for scanning electron microscopy (SEM) and transmission electron microscopy (TEM). The spherical AgNPs were observed by SEM and TEM to have an average size of 45 nm (Figure 3a) and 50 nm (Figure 3b), and exhibited a surface plasmon resonance absorption peak at 400 nm. For the Apt-ACT-CD_NAg2_-H_2_O_2_-DBD-AgNPs system, when there was no ACT in the system, Apt encapsulated the carbon dots, inhibiting the CD_N_-catalyzed H_2_O_2_-DBD reaction and forming fewer DBD_ox_ probes, which caused weak AgNPs aggregations with an average particle size of 50 nm (Figure 3b). With the increase of ACT concentration, CD_N_ was gradually released and the catalytic effect was enhanced to form more DBD_ox_ probes, which caused strong AgNPs aggregations with an average particle size of 60 nm (Figure 3c). In theory, the sizes of the AgNPs, the blank, and the analytical systems are the same. The size difference was ascribed to the different conditions of the three systems. The prepared CD_N_ were diluted and dropped onto a silicon wafer for transmission electron microscopy scanning. The average particle diameter was about 20 nm (Figure 3d). The energy spectra (Figure 3e) indicated that there were Ag atoms doped in the CD because the peak at 3.0 keV was ascribed to Ag. For the CD infrared spectra (Figure 3f) eight peaks were observed, at 3908, 3785, 3430, and 3202 cm^−1^ ascribing to O–H stretching vibrations, 1716 cm^−1^ ascribing to C=O stretching vibration, 1666 cm^−1^ ascribing to C=C conjugate structure stretching vibrations, 1384 cm^−1^ ascribing to an N=N stretching motion, 1077 cm^−1^ ascribing to C–H in-plane bending vibrations, and at 642 cm^−1^ ascribing to N–C=O bending vibrations.

### 3.5. Optimization of the Analytical Conditions

The effect of the concentration of H_2_O_2_, DBD, CD_NAg2_, pH, Apt and AgNPs, the reaction temperature and the time on the SERS signal of the system were investigated, respectively (Figure 4). The SERS signal was the strongest when 0.08 mmol/L H_2_O_2_, 0.025 mmol/L DBD, 6.67 μg/L CD_NAg2_, pH 3.6 HAc-NaAc buffer solution, 0.026 μmol/L Apt and 0.11 mmol/L AgNPs solution was added. Thus, those concentrations were chosen for use, respectively. When the reaction temperature was 50 °C for 20 min, the SERS was the strongest, so these conditions were selected.

### 3.6. Influence of Interfering Ions

The interference of coexisting ions (CES) on the Apt-ACT-CD_N_-H_2_O_2_-DBD-AgNPs SERS system was determined by 0.1 μg/L ACT. The results showed that 1000 times Zn^2+^, Ca^2+^, Ni^2+^, Mn^2+^, Co^2+^, NH_4_Cl and CO_3_^2−^, 500 times Ba^2+^, Mg^2+^, K^+^, HCO_3_^−^ and NO_2_^−^, 250 times Fe^3+^, Bi^3+^, Cu^2+^, Pb^2+^, Al^3+^ and Hg^2+^, 250 times imidacloprid, chlorpyrifos and atrazine, 100 times Cr^6+^, Fe^2+^, and carbendazim had no interference on the determination of ACT (Table 2). Therefore, the SERS method had acceptable selectivity.

### 3.7. Working Curve

Under the selected conditions, the relationship between the ACT concentration (C) and its corresponding Δ*I* was obtained (Figure 5, Table 3). In the four systems, the slope of the CD_NAg2_ system was the largest due to CD_NAg2_ having the strongest catalysis. Therefore, the system was chosen for the assay of ACT in real samples. This SERS method is simpler and more sensitive than the reported spectral method for determining ACT [28,29,30,31,32,33,34,35,36,37] (Table 4).

### 3.8. Sample Analysis

Pakchoi, cucumber and tomato were purchased from Guilin Agricultural Market and 50 g samples were weighed. The grinding bowl was ground thoroughly and 1 mL of 99.5% acetone was added. The mixture was filtered with filter paper and centrifuged at 1000 RPM for 5 min. The supernatant was collected and stored in a refrigerator at 4 °C for use. We followed the procedure for SERS detection and tested the recovery. Table 5 indicates that the recovery was 95.7–99.4% and the relative standard deviation was 3.5–5.6%.

## 4. Conclusions

In this paper, a nitrogen/silver co-doped carbon dot with high stability and high catalytic activity was synthesized by microwave method. Based on the catalytic effect of the carbon dot on the H_2_O_2_-DBD SERS reaction, combined with the specific reaction of ACT-Apt, a new SERS quantitative analysis method for ACT was constructed. It has the advantages of simple operation, high sensitivity and good selectivity.

## Figures and Tables

**Figure 1 nanomaterials-09-00480-f001:**
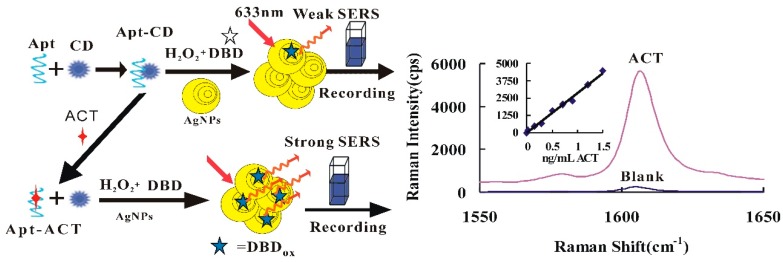
Carbon dot (CD) catalytic amplification surface-enhanced Raman scattering (SERS) analytical platform based on aptamer with 3,3′-dimethylbiphenyl-4,4′-diamine (DBD) oxidizing reaction.

**Figure 2 nanomaterials-09-00480-f002:**
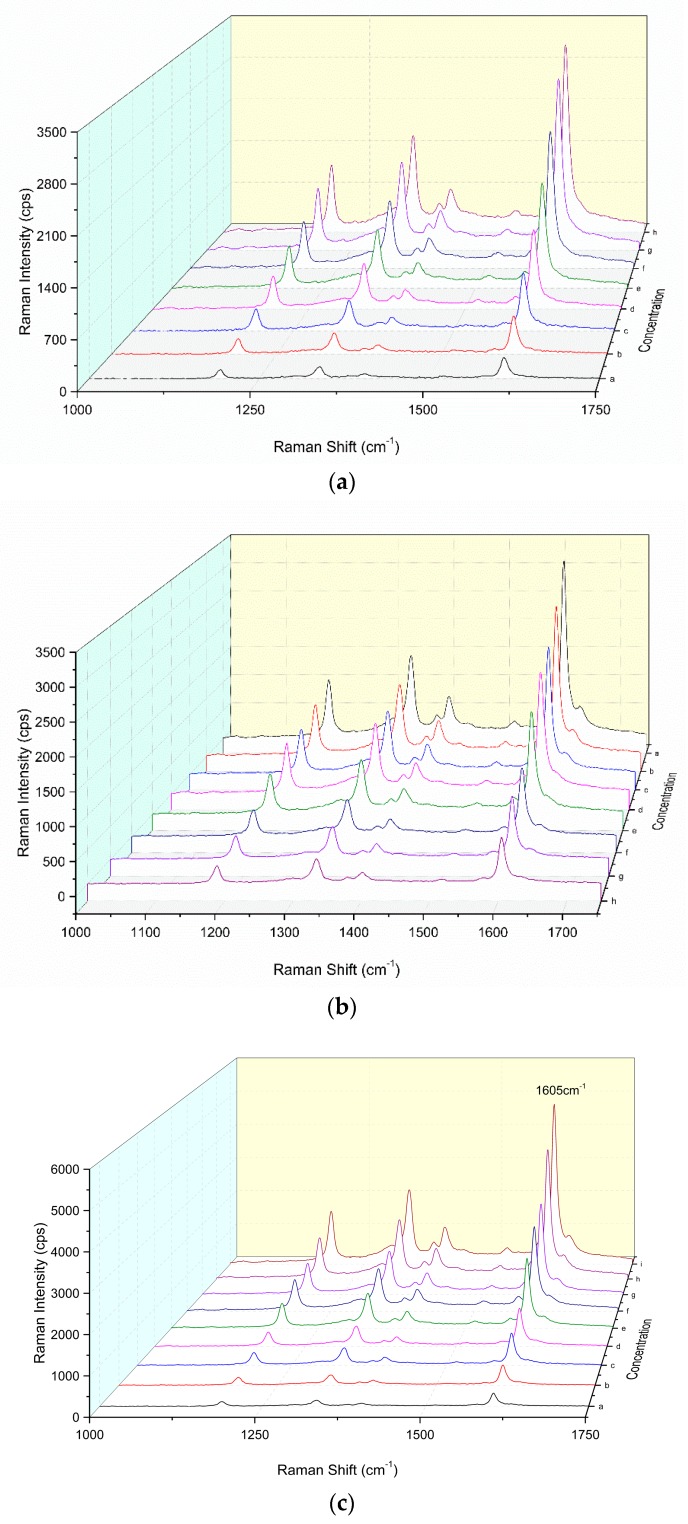
SERS spectra of a CD_NAg2_-H_2_O_2_-DBD-AgNPs system. (**a**) 0.13 mmol/L H_2_O_2_ + 33 μmol/L DBD + 0.11 mmol/L AgNPs, a–h were 0, 0.011, 0.057, 0.113, 0.17, 0.227, 0.34, and 0.45 mg/L CD_NAg2_, respectively. (**b**) 0.45 mg/L CD_NAg2_ + 0.13 mmol/L H_2_O_2_ + 33 μmol/L DBD + 0.11 mmol/L AgNPs, the a–h were 0, 3.1, 7.2, 15.5, 20.7, 25.8, 31, and 36.2 nmol/L aptamer (Apt). (**c**) 25.8 nmol/L Apt + 0.45 mg/L CD_NAg2_ + 0.08 mmol/L H_2_O_2_ + 0.025 mmol/L DBD + 0.11 mmol/L AgNPs, a–i were 0, 0.03, 0.15, 0.3, 0.5, 0.7, 0.9, 1.2, and 1.5 μg/L acetamiprid (ACT), respectively.

**Figure 3 nanomaterials-09-00480-f003:**
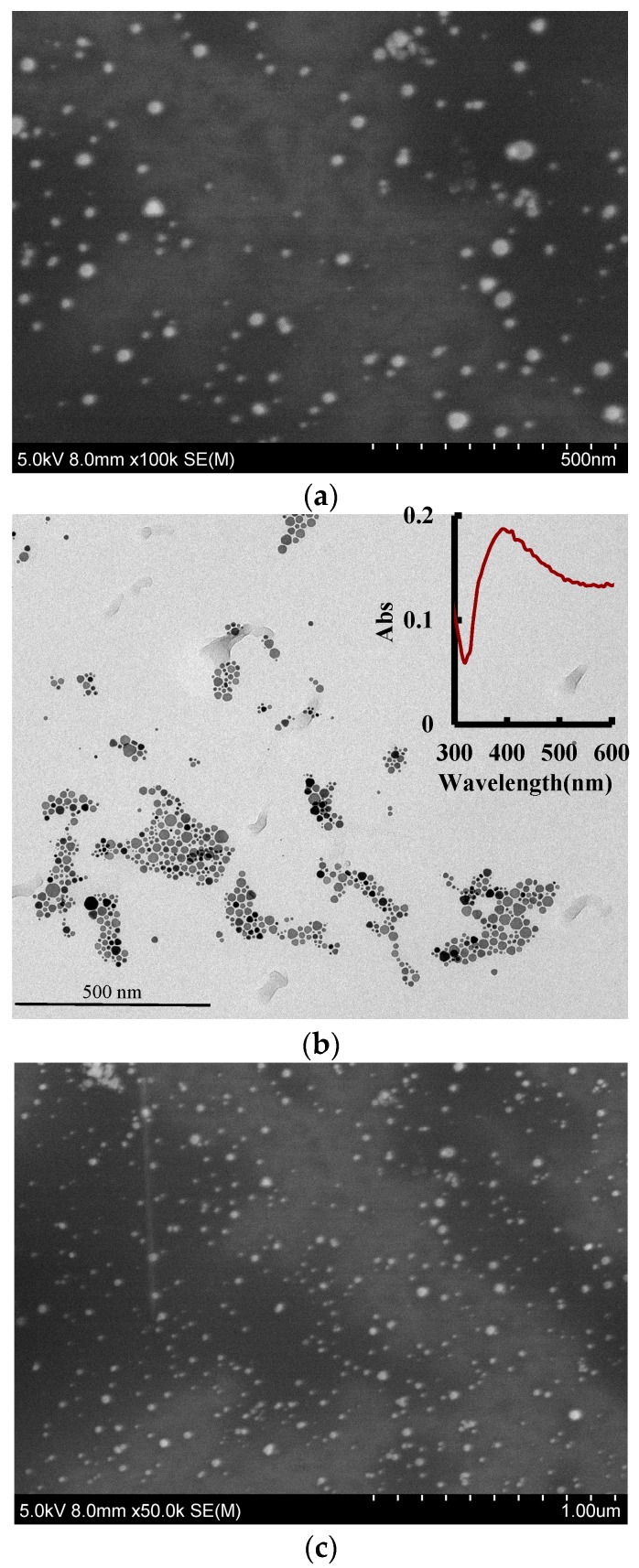
Scanning electron microscopy (SEM), transmission electron microscopy (TEM), energy spectra and infrared spectra. (**a**) SEM image of AgNPs; (**b**) TEM image of AgNPs (the inserted Figure was the surface plasmon resonance absorption spectrum of AgNPs); (**c**) 0.026 μmol/L Apt + 6.67 μg/L CD_NAg2_ + 0.08 mmol/L H_2_O_2_ + 0.025 mmol/L DBD + pH 3.6 HAc-NaAc + 0.11 mmol/L AgNPs; (**d**) SEM of c + 0.5 μg/L ACT; (**e**) TEM of CD_NAg2_; (**f**) energy spectra of CD_NAg2_; and (**g**) infrared spectra (IR) image of CD_NAg2_.

**Figure 4 nanomaterials-09-00480-f004:**
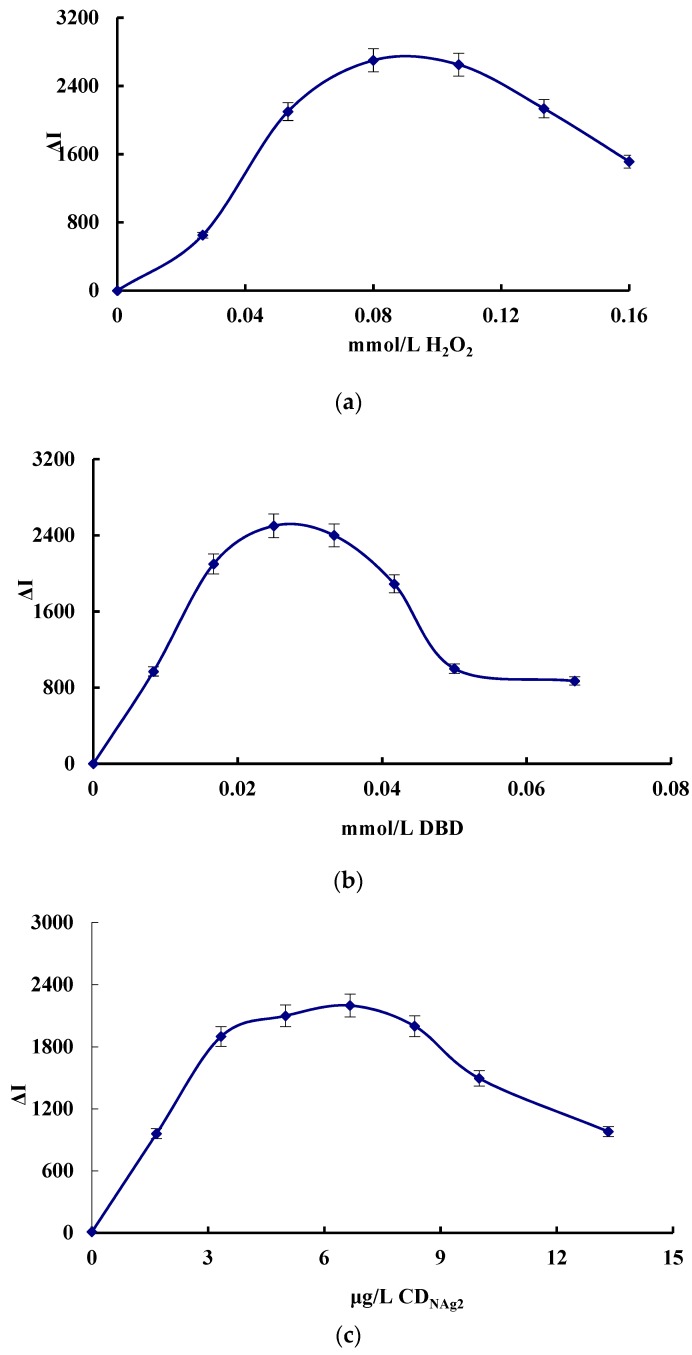
Effect of reagent concentration, reaction temperature and time on Δ*I*. (**a**) H_2_O_2_, (**b**) DBD, (**c**) CD_NAg2_, (**d**) Apt, (**e**) AgNPs, (**f**) pH, (**g**) reaction temperature, and (**h**) reaction time.

**Figure 5 nanomaterials-09-00480-f005:**
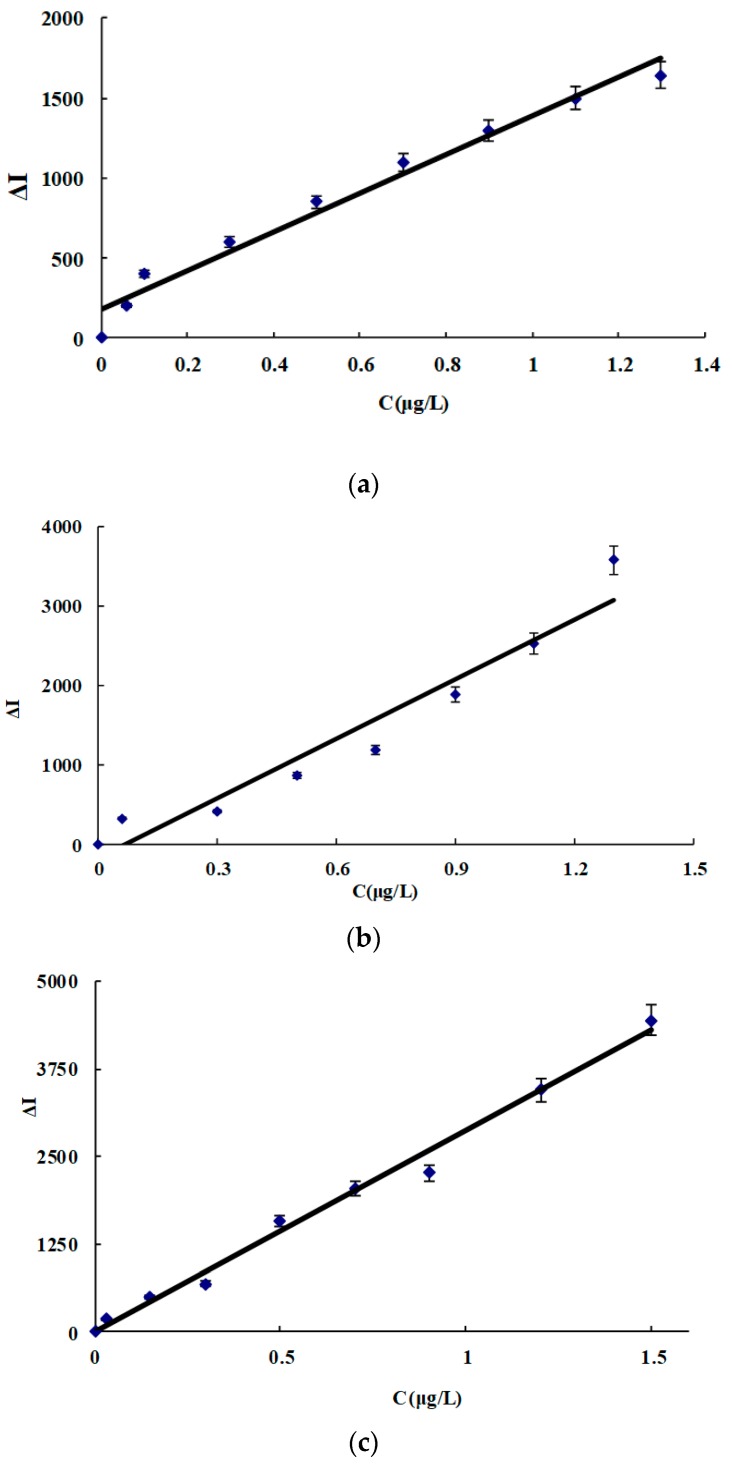
Working curve for ACT. (**a**) CD_N_ system, (**b**) CD_NAg1_ system, (**c**) CD_NAg2_ system, and (**d**) CD_NAg3_ system.

**Table 1 nanomaterials-09-00480-t001:** Nanocatalysis and Apt inhibition.

Nanocatalytic System	Dynamic Range	Regress Equation	Coefficient
CD_N_	1.7–13 µg/L CD	Δ*I* = 610*C* − 37	0.9058
CD_NAg1_	11–450 µg/L CD	Δ*I* = 6102*C* + 180	0.9886
CD_NAg2_	8–120 µg/L CD	Δ*I* = 8180*C* + 16	0.979
CD_NAg3_	9–260 µg/L CD	Δ*I* = 7870*C* + 238	0.9198
Apt-CD_N0_	3–26 nmol/L Apt	Δ*I* = 49*C* + 125	0.9307
Apt-CD_NAg1_	5–30 nmol/L Apt	Δ*I* = 58*C* − 95	0.9387
Apt-CD_NAg2_	3–36 nmol/L Apt	Δ*I* = 60*C* + 270	0.9598
Apt-CD_NAg3_	10–50 nmol/L Apt	Δ*I* = 42*C* − 130	0.9508

**Table 2 nanomaterials-09-00480-t002:** Effect of coexisting ions (CES).

CES	Tolerance (*C*_ACT_/*C*c_ES_)	Error (%)	CES	Tolerance (*C*_ACT_/*C*c_ES_)	Error (%)
Zn^2+^	1000	3.0	Fe^3+^	250	6.5
Ca^2+^	1000	−7.0	Bi^3+^	250	2.9
Ni^2+^	1000	5.0	Cu^2+^	250	4.5
Mn^2+^	1000	3.0	Pb^2+^	250	5.0
Co^2+^	1000	4.7	Al^3+^	250	6.0
NH_4_Cl	1000	4.8	Cr^6+^	100	6.0
CO_3_^2−^	1000	6.5	Fe^2+^	100	8.0
Ba^2+^	500	−5.4	Hg^2+^	250	−5.0
Mg^2+^	500	−5.1	imidacloprid	200	6.0
K^+^	500	−4.2	atrazine	200	3.0
HCO_3_^−^	500	5.2	carbendazim	100	7.0
NO_2_^−^	400	−4.0	chlorpyrifos	200	4.3

**Table 3 nanomaterials-09-00480-t003:** Analytical characteristics of the carbon dot (CD) catalytic amplification surface-enhanced Raman scattering (SERS) assay for acetamiprid (ACT).

System	Linear Range (μg/L)	Regress Equation	Coefficient	DL (μg/L)
CD_N_	0.06−1.3	Δ*I* = 1240*C* + 180	0.9754	0.026
CD_NAg1_	0.06−1.3	Δ*I* = 2500*C* − 170	0.9367	0.029
CD_NAg2_	0.01−1.5	Δ*I* = 2800*C* + 20	0.9967	0.006
CD_NAg3_	0.06−0.9	Δ*I* = 1650*C* + 50	0.9832	0.027

**Table 4 nanomaterials-09-00480-t004:** Comparison of some reported molecular spectral methods for ACT.

Method	Principle	Linear Range	Detection Limt	Comment	Ref.
Fluorescence	The aptasensor for ACT based on the inner filter effect between nanogold and CD.	5–100 μg/L	1.08 μg/L	High precision but low sensitivity.	[34]
Fluorescence	A label-free triplex-to-G-quadruplex molecular switch for ACT.	10–500 nM	2.38 nM	Fast but low sensitivity.	[35]
Resonance Light Scattering	Nanogold modified Apt-based resonance light scattering for ACT.	0–100 nM	1.2 nM	High sensitivity but complicated operation.	[36]
Colorimetry	Aptamer-based colorimetric sensing of ACT in soil samples based on aggregation of Au nanoparticles.	0.075–7.5 μM	5 nM	Fast and selective but low sensitivity.	[25]
SERS	Gold on carbon fiber needle-like SERS substrate for ACT by combining with thin-layer chromatography.	0.1–10 μg/mL	0.05 μg/mL	Selective but low sensitivity.	[37]
DBD_ox_ SERS probe	Apt used to modulate CD catalysis to generate DBD_ox_ to detect ACT.	0.01–1.5 μg/mL	0.006 μg/L	High sensitivity and good selectivity.	This method

**Table 5 nanomaterials-09-00480-t005:** Determination results of ACT in water samples.

Sample	Content (μg/L)	Added (μg/L)	Found (μg/L)	Recovery (%)	Relative Standard Deviation (%)	Ref. Results (μg/L)
Pakchoi	—	0.1	0. 0960	96.0	5.0	—
Cucumber	0.16	0.1	0.0957	95.7	3.5	0.15
Tomato	0.10	0.1	0.0995	99.5	5.6	0.12

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
