# Peer review of "Doped N/Ag Carbon Dot Catalytic Amplification SERS Strategy for Acetamiprid Coupled Aptamer with 3,3′-Dimethylbiphenyl-4,4′-diamine Oxidizing Reaction"

_nanomaterials, 2019, doi:10.3390/nano9030480_

Reviewer 1 Report

I recommend the revision of the manuscript considering the following issues. 

1. The language in the manuscript needs to be improved, there are quite a few language typos, errors, and mistakes. 

2. For carbon dots based fluorescence sensing, it is necessary to add some excellent review works in recent years. 

3. For SERS based sensing, the repeatability and reproducibility are the challenges. It is necessary to add the averages and standard deviations of SERS signal results from several times experiments. 

4. For Figure 4, X-axis is missing for several figures.

Author Response

I recommend the revision of the manuscript considering the following issues. 

1. The language in the manuscript needs to be improved, there are quite a few language typos, errors, and mistakes.

Answer: Thanks! The language in the manuscript was improved carefully, and the language typos, errors and mistakes were corrected.

2. For carbon dots based fluorescence sensing, it is necessary to add some excellent review works in recent years. 

Answer: Recent important two reviews about carbon dots based fluorescence sensing (14. M. L. Liu, B. B. Chen, C. M. Li, C. Z. Huang, Carbon dots: synthesis, formation mechanism, fluorescence origin and sensing applications. Green Chem. 2019, 21, 449-471; 15. T. S. Atabaev, Doped carbon dots for sensing and bioimaging applications: A minireview. Nanomaterials 2018, 8, 342) were added.

3. For SERS based sensing, the repeatability and reproducibility are the challenges. It is necessary to add the averages and standard deviations of SERS signal results from several times experiments.

Answer: The standard deviations of SERS signal results for three times experiments was added in the Figure 4.

4. For Figure 4, X-axis is missing for several figures.

Answer: The “a: H2O2; b: DBD; c: CDNAg2; d: Apt; e: AgNPs; f: pH; g: reaction temperature; h: reaction time.” was added.

Reviewer 2 Report

Comments to the Author

The manuscript “A doped-N/Ag carbon dot catalytic amplification SERS strategy for acetamiprid coupled aptamer with 3,3'-dimethylbiphenyl-4,4'-diamine oxidizing reaction” by Feng et al., presents a detection of an insecticide acetamiprid coupled with a specific aptamer by a SERS coupled with a doped-N-Ag carbon dot.

This paper presents a potentially interesting work that show optimized acetamiprid detection but I can recommend it to be published in Nanomaterials only after several modifications:

1)      Abbreviations given in the Abstract should be given in the main text too (as DBD, ACT). The main text is difficult to follow because there are too many abbreviations.  As the main text is not too long it will be better to keep some full names through the whole manuscript. In case one word is repeated less than 3 times in the text no abbreviation is needed.

2)      How the aptamer used in the work was selected? If any previous publication on it, it should be cited.

3)      Insert in Fig. 2C should be explained in the legend

4)      Figure 4 should be rescaled since les graphs are not completely visible.

5)      Part 3.5 should be re-written because it contains too many repetitions;

6)      The calibration curves enabled to calculate analytical parameters in Table 3 should be presented. How many concentrations were tested to calculate these parameters?

7)      Finally, the main finding in this paper is the amplification of the signal read out by the doped-N/Ag carbon dots. To estimate the amplification it is necessary to present analytical parameters of the SERS measurement without carbon dots. In the present form of the paper, it is not clear what signal amplification was reached by coupling SERS with doped-N/Ag carbon dots.

Author Response

The manuscript “A doped-N/Ag carbon dot catalytic amplification SERS strategy for acetamiprid coupled aptamer with 3,3'-dimethylbiphenyl-4,4'-diamine oxidizing reaction” by Feng et al., presents a detection of an insecticide acetamiprid coupled with a specific aptamer by a SERS coupled with a doped-N-Ag carbon dot.

This paper presents a potentially interesting work that show optimized acetamiprid detection but I can recommend it to be published in Nanomaterials only after several modifications:

1)  Abbreviations given in the Abstract should be given in the main text too (as DBD, ACT). The main text is difficult to follow because there are too many abbreviations.  As the main text is not too long it will be better to keep some full names through the whole manuscript. In case one word is repeated less than 3 times in the text no abbreviation is needed.   

Answer: Thanks! The abbreviations such as DBD and ACT repeated more than 3 times were given in the main text.

2)  How the aptamer used in the work was selected? If any previous publication on it, it should be cited.

Answer: A reference [35] was cited in the line 93 of Introduction section.

3)  Insert in Fig. 2C should be explained in the legend

Answer: The inserted figure was moved to Figure 5c and explained.

4)  Figure 4 should be rescaled since les graphs are not completely visible.

Answer: The Figure 4 was rescaled.

5)  Part 3.5 should be re-written because it contains too many repetitions;

Answer: The part was revised according to the comments.

6) The calibration curves enabled to calculate analytical parameters in Table 3 should be presented. How many concentrations were tested to calculate these parameters?

Answer: The calibration curves were added (Figure 5), 6-9 concentrations were tested to calculate these parameters.

7)  Finally, the main finding in this paper is the amplification of the signal read out by the doped-N/Ag carbon dots. To estimate the amplification it is necessary to present analytical parameters of the SERS measurement without carbon dots. In the present form of the paper, it is not clear what signal amplification was reached by coupling SERS with doped-N/Ag carbon dots.

Answer: Thanks! We also considered this problem that the both ACT-Apt and ACT-Apt –CDNAg2 systems have no SERS signal if the Apt reaction can not be coupled the catalytic reaction. That is, the SERS signal come from the product of the catalytic reaction.

Reviewer 3 Report

the mauscript is very difficult to read, as english language is used in a rough way, with sentences lacking sometimes the subjects, sometimes the verb, sometimes the conjunctions, and sometime any meaning at all.

Despite this, the work looksas having something  interesting: with a careful editing and rewriting probably also the reviewing process could be performed more easily...

just to make few examples: page 7, line 240: how is fluorescence involved?

or the phrase

"Nanoparticles have novel optical properties, such as surface  plasmon resonance (SPR) effects, and SPR-based highly enhanced surface-enhanced Raman  scattering (SERS) technology, which is the most important application of nanoparticles in Raman  spectroscopy, would be improved sensitivity greatly." has no meaning in this form.

Moreover, the introduction is badly arraanged: the term SERS is used before explaining the acronym meaning, and no accounts on SERS related issues are reported (see for example

Nanotechnology 2019, 30 (2), 025302)

Going to figures, for example, they  are mostly unreadable:

figure 1 there is no evidence of linearity cited in line 156: just one SERS spectra is reported... of course the linearity can be observed in figure 2, but also figure 2 is almost unreadable

figure 2: caption is too complex an figure usless in this form

inset in figure 2 c is unreadable and useless

it is not clear what we see in figure 3 a nd b: carbon dots? AgNP? both together?

A characterization of AgNP alone is necessary (see following comments)

figure 4 completely unreadable, x axis missed in most graphs,

Also, I am very interested about the morphology of Ag colloid: a blue turning to orange red silver colloid should have some degree of anisotrpy, or dimensions quite larger (>40 nm), while  standard silver sol of spherical shape an dimensions of few nm should be yellow, witha LSPR peak close to 400 nm

is there any characterization of the Ag nanoparticles? the preparation as described

"Preparation of the nanosilver sol (AgNPs): A 44mL water was added into a conical flask, then 2 mL 125 10mM AgNO3, 2.0mL 100 mM trisodium citrate, 600μL 30% H2O2, 600μL 0.1M NaBH4 were added into the water successively. After the color turn into blue with rapidly stirring, the mixture was shift to the air light wave stove immediately. After irradiation for 10 min at 250℃, the color went from blue to orange red"

should in my opinion give large and prismatic nano-objecs, as citrate promote anisotropic growth, while H2O2 selective etching of smaller NPs.

This is not a trivial point, as, as authors surely know, SERS response is strongly affected by nanoparticles morphology and dimension, so, to make the method reproducible, authors should give a precise characterization of Ag NP.

these are only few of the aspect that should be corrected.

nevertheless, I think that some intersting results are present, even if they are hindered by the confuse presentation and numerous mistakes, so my idea is to  reject the paper in the present form but in the same time encouraging resubmission after an hard rewriting, involving reorganization of intro and of data presentation, adding AgNP characterization and a deep language editing

Author Response

the mauscript is very difficult to read, as English language is used in a rough way, with sentences lacking sometimes the subjects, sometimes the verb, sometimes the conjunctions, and sometime any meaning at all.

Answer: English language was improved carefully.

Despite this, the work looks as having something interesting: with a careful editing and rewriting probably also the reviewing process could be performed more easily...

just to make few examples: page 7, line 240: how is fluorescence involved?

Answer: It was an error, and was revised.

or the phrase

"Nanoparticles have novel optical properties, such as surface plasmon resonance (SPR) effects, and SPR-based highly enhanced surface-enhanced Raman scattering (SERS) technology, which is the most important application of nanoparticles in Raman spectroscopy, would be improved sensitivity greatly." has no meaning in this form.

Answer: The “Nanoparticles have novel optical properties, such as surface plasmon resonance effects, and SPR-based highly enhanced surface-enhanced Raman scattering (SERS) technology, which is the most important application of nanoparticles in Raman spectroscopy, would be improved sensitivity greatly. It also has real-time,” was changed to “Surface-enhanced Raman scattering (SERS) has real-time,”.

Moreover, the introduction is badly arraanged: the term SERS is used before explaining the acronym meaning, and no accounts on SERS related issues are reported (see for example Nanotechnology 2019, 30 (2), 025302)

Answer: Thanks! The introduction was revised, the acronym was given before the term SERS. The “Although many SERS studies have been reported, most of them are qualitative detection. As an analytical technique, highly sensitive and selective SERS quantitative analytical method is also important. This depends on reproducible SERS substrates and highly selective biochemical reactions such as Apt reaction.” was added in the first graph of Introduction section.

Going to figures, for example, they are mostly unreadable:

figure 1 there is no evidence of linearity cited in line 156: just one SERS spectra is reported... of course the linearity can be observed in figure 2, but also figure 2 is almost unreadable

figure 2: caption is too complex an figure useless in this form

inset in figure 2 c is unreadable and useless

Answer: Figure 1 was revised, and the linear curve was added. Figure 2 caption was revised, it was simple, and the inserted figure was moved to Figure 5c.

it is not clear what we see in figure 3 a and b: carbon dots? AgNP? both together?

A characterization of AgNP alone is necessary (see following comments)

Answer: The graphs were enlarged to see clearly. The SEM of AgNP was added in Figure 3a.

figure 4 completely unreadable, x axis missed in most graphs,

Answer: The Figure 4 was revised to see clearly.

Also, I am very interested about the morphology of Ag colloid: a blue turning to orange red silver colloid should have some degree of anisotrpy, or dimensions quite larger (>40 nm), while standard silver sol of spherical shape an dimensions of few nm should be yellow, with a LSPR peak close to 400 nm

is there any characterization of the Ag nanoparticles? the preparation as described

Answer: A reference [37] was cited in the section of 2.2, the reference was given the size, color (yellow) of the AgNPs.

"Preparation of the nanosilver sol (AgNPs): A 44mL water was added into a conical flask, then 2 mL 125 10mM AgNO3, 2.0mL 100 mM trisodium citrate, 600μL 30% H2O2, 600μL 0.1M NaBH4 were added into the water successively. After the color turn into blue with rapidly stirring, the mixture was shift to the air light wave stove immediately. After irradiation for 10 min at 250℃, the color went from blue to orange red"

should in my opinion give large and prismatic nano-objecs, as citrate promote anisotropic growth, while H2O2 selective etching of smaller NPs.

This is not a trivial point, as, as authors surely know, SERS response is strongly affected by nanoparticles morphology and dimension, so, to make the method reproducible, authors should give a precise characterization of Ag NP.

Answer: A reference [37] was cited in the section of 2.2. In this procedure for preparation of AgNPs, NaBH4 was a strong reducer that reduced AgNO3 to produce very small AgNPs as catalyst. It catalyzed the reaction of H2O2-AgNO3 to produce AgNPs, in which H2O2 is a reducer. In another work of our research group (Guiqing Wen, Yanghe Luo, Aihui Liang*, Zhiliang Jiang*. Autocatalytic oxidization of nanosilver and its application to spectral analysis. Scientific Reports, 2014, 4: 3990; Zhiliang Jiang#,*, Guiqing Wen#, Yanghe Luo, Xinghui Zhang, Qingye Liu, Aihui Liang*. A new silver nanorod SPR probe for detection of trace benzoyl peroxide. Scientific Reports, 2014, 4: 5323), the nanoparticles morphology and dimension for AgNPs were studied in details and used in analysis. The SEM of AgNPs was added in Figure 3a.

these are only few of the aspect that should be corrected.

nevertheless, I think that some interesting results are present, even if they are hindered by the confuse presentation and numerous mistakes, so my idea is to reject the paper in the present form but in the same time encouraging resubmission after an hard rewriting, involving reorganization of intro and of data presentation, adding AgNP characterization and a deep language editing

Answer: The manuscript was revised carefully, the first paragraph of section Introduction was re-arranged, and the AgNP characterization by SEM was added.

Round  2

Reviewer 2 Report

The authors improved the manuscript in many aspects. 

Author Response

The authors improved the manuscript in many aspects.

Answer: Thanks! The manuscript was improved in red letters.

Reviewer 3 Report

the NP characterization is still a little bit confused... which is the size of AgNP? why it moves from 45 to 55 to 60 nm? is it just because of magnification or different conditions of images?

authors should provide, maybe in the supplementary material, the LSPR spectra of AgNP used in the work and maybe some better TEM. I have looked at the cited article, in my opinion the AgNP are small plates, and not spherical. This is not fundamental, nevertheless a complete characterization of AgNP must be showed for an analytical reproducible method, so at least the UV-vis spectra showing the LSPR band of AgNP must be reported

In the introduction, some reference on SERS need for reproducibility should be given, after the paraghraph added in the revison, for example Nanotechnology 2019, 30 (2), 025302 and Ana. Bioanal. Chem. 394 1729

Finally, in my opinion the language is still far from being perfect, anyway a quite giant leap has been performed from the first version.

Author Response

the NP characterization is still a little bit confused... which is the size of AgNP? why it moves from 45 to 55 to 60 nm? is it just because of magnification or different conditions of images?

Answer: In theory, the sizes of the AgNPs, the blank and the analytical systems are same. The size different was ascribing to the different conditions of the three systems. This was added in the section of 3.4.

authors should provide, maybe in the supplementary material, the LSPR spectra of AgNP used in the work and maybe some better TEM. I have looked at the cited article, in my opinion the AgNP are small plates, and not spherical. This is not fundamental, nevertheless a complete characterization of AgNP must be showed for an analytical reproducible method, so at least the UV-vis spectra showing the LSPR band of AgNP must be reported

Answer: Thanks! The used AgNPs in orange red were prepared according to reference [37] by light wave procedure. The TEM and surface plasmon resonance absorption spectra were added in the Figure 3b.

In the introduction, some reference on SERS need for reproducibility should be given, after the paraghraph added in the revison, for example Nanotechnology 2019, 30 (2), 025302 and Ana. Bioanal. Chem. 394 1729

Answer: In the introduction, the “Reproducibility not only affects qualitative analysis, but is fatal to quantitative analysis, which limits its development. To overcome the shortage, some reproducible preparations of clean and highly Ag and Au SERS active substrates were reported [14]. Bassi et al [15] prepared a recyclable SERS active glass chips, the gold nanostars were grafted on functionalized glasses by means of electrostatic interactions and then they were coated with a silica layer of controllable thickness. The SERS activity were examined in terms of reproducibility using rhodamine 6G molecular probes.” were added in the revision.

[14] Lin, X.M.; Cui, Y.; Xu, Y.H.; Ren, B.; Tian, Z.Q. Surface-enhanced Raman spectroscopy: substrate-related issues. Anal. Bioanal. Chem. 2009, 394, 1729-1749

[15] Bassi, B.;

Finally, in my opinion the language is still far from being perfect, anyway a quite giant leap has been performed from the first version.

Answer: The manuscript was improved carefully in red letters.

Round  3

Reviewer 3 Report

figure 3 caption is wrong and incomplete: figure 3 b should be a TEM and not a SEM, and inset in figure 3b must be described in the caption

also in line 210 probably there is a mistake, as dimensions derived from TEM should refre to figure 3b and not 3a

please check carefully the manuscript once again.

Author Response

Reviewer 3

figure 3 caption is wrong and incomplete: figure 3 b should be a TEM and not a SEM, and inset in figure 3b must be described in the caption

Answer: In figure 3 b, the “SEM” was changed to “TEM”, and the “and the inserted Figure was the surface plasmon resonance absorption spectrum of AgNPs” was described in the caption.

also in line 210 probably there is a mistake, as dimensions derived from TEM should refre to figure 3b and not 3a

Answer: The “Figure 3a” was changed to “Figure 3b” in the line 209.

please check carefully the manuscript once again.

Answer: The manuscript was improved carefully in red letters.

Round  4

Reviewer 3 Report

authors have revised the paper correctly and it can be published.